# 3D Chromatin Architecture Re-Wiring at the *CDH3/CDH1* Loci Contributes to E-Cadherin to P-Cadherin Expression Switch in Gastric Cancer

**DOI:** 10.3390/biology12060803

**Published:** 2023-05-31

**Authors:** Celina São José, Carla Pereira, Marta Ferreira, Ana André, Hugo Osório, Irene Gullo, Fátima Carneiro, Carla Oliveira

**Affiliations:** 1Instituto de Investigação e Inovação em Saúde, Universidade do Porto, 4200-135 Porto, Portugal; cjose@ipatimup.pt (C.S.J.); hosorio@i3s.up.pt (H.O.); igullo@ipatimup.pt (I.G.);; 2Institute of Molecular Pathology and Immunology of the University of Porto, 4200-135 Porto, Portugal; 3Doctoral Programme in Biomedicine, Faculty of Medicine, University of Porto, 4200-319 Porto, Portugal; 4Doctoral Program in Computer Sciences, Faculty of Sciences, University of Porto, 4169-007 Porto, Portugal; 5Faculty of Medicine, University of Porto, 4200-319 Porto, Portugal; 6Department of Pathology, Centro Hospitalar Universitário São João, 4200-319 Porto, Portugal

**Keywords:** gastric cancer, cadherins switch, regulatory elements, 3D chromatin architecture, chromatin interactions, CDH1, CDH3

## Abstract

**Simple Summary:**

The cell architecture is created and maintained via adhesion molecules, called cadherins, that keep similar cells bound to each other. E-cadherin is a cadherin encoded by the *CDH1* gene, essential in normal epithelia. P-cadherin, encoded by the *CDH3* gene, exerts a similar function but is mainly present in tumours. E- to P-cadherin switch is a common event in several epithelial tumours. We unveiled a mechanism explaining the E- to P-cadherin switch (*CDH1* to *CDH3* switch), which we found to occur in many gastric cancers. We induced this switch by depleting *CDH1* in gastric cancer cells, and, as they started to produce *CDH3*, this rescued cell-adhesion. Despite this rescue, cells bearing the switch showed migratory and proliferative features, commonly observed in aggressive tumours. We observed that *CDH1* depletion leads to novel genomic interactions between the *CDH1* and *CDH3* gene sequences. We identified the sequences needed for these interactions to occur, which seem also responsible for controlling the amount of E- or P-cadherin molecules in the cell. Our data provide evidence that loss of E-cadherin is able to compromise the normal interactions between the *CDH1* to *CDH3* gene sequences, allowing the expression of P-cadherin in gastric cancers.

**Abstract:**

Cadherins are cell–cell adhesion molecules, fundamental for cell architecture and polarity. E-cadherin to P-cadherin switch can rescue adherens junctions in epithelial tumours. Herein, we disclose a mechanism for E-cadherin to P-cadherin switch in gastric cancers. *CDH1* and *CDH3* mRNA expression was obtained from 42 gastric tumours’ RNA-seq data. CRISPR-Cas9 was used to knock out *CDH1* and a putative regulatory element. *CDH1*-depleted and parental cells were submitted to proteomics and enrichment GO terms analysis; ATAC-seq/4C-seq with a *CDH1* promoter viewpoint to assess chromatin accessibility and conformation; and RT-PCR/flow cytometry to assess *CDH1*/E-cadherin and *CDH3*/P-cadherin expression. In 42% of gastric tumours analysed, *CDH1* to *CDH3* switch was observed. *CDH1* knockout triggered *CDH1*/E-cadherin complete loss and *CDH3*/P-cadherin expression increase at plasma membrane. This switch, likely rescuing adherens junctions, increased cell migration/proliferation, commonly observed in aggressive tumours. E- to P-cadherin switch accompanied increased *CDH1* promoter interactions with *CDH3*–eQTL, absent in normal stomach and parental cells. *CDH3*–eQTL deletion promotes *CDH3*/*CDH1* reduced expression. These data provide evidence that loss of *CDH1*/E-cadherin expression alters the *CDH3* locus chromatin conformation, allowing a *CDH1* promoter interaction with a *CDH3*-eQTL, and promoting *CDH3*/P-cadherin expression. These data highlight a novel mechanism triggering E- to P-cadherin switch in gastric cancer.

## 1. Introduction

Cadherins are tissue-specific transmembrane glycoproteins that play a key role in stable cell–cell adhesion, cellular polarization, maintenance of tissue architecture, and embryogenesis [1,2]. E-cadherin and P-cadherin are encoded by *CDH1* and *CDH3* genes, respectively, and present a great degree of sequence and gene structure homology [3]. Despite their homology, both proteins display tissue-specific expression patterns and functions, being often mutually exclusive at the cellular level [3]. While E-cadherin is widely expressed in epithelial tissues [4], P-cadherin expression is restricted to organ regions, such as myoepithelial cells of the breast or mesothelia [5]. Whereas E-cadherin acts as a tumour suppressor in most epithelial tissues, P-cadherin de novo expression or upregulation has been linked to mammary cells’ proliferation and migration in mice [6], tumour-promotion in gastric [7,8] and breast cancer [9,10], and worse prognosis in breast cancer [11,12,13].

Aberrant cadherin isoform switching occurs in tumours with implications for patients’ survival [14,15]. E-cadherin to N-cadherin switch is a mark of epithelial to mesenchymal transition, in which differentiated epithelial cells, expressing E-cadherin, transition to dedifferentiated mesenchymal-like cells, expressing N-cadherin, and are prone to invade and metastasize [16]. Evidence of E-cadherin to P-cadherin switching has been observed in both gastric [17] and breast cancer, with implications for survival in the latter [11].

Mechanisms for E-cadherin to P-cadherin switching have been suggested, mainly through MiR-34a/WNT1 cascade [18], but fail to explain all observed switching phenotypes. As *CDH1* and *CDH3* are juxtaposed genes, derived from gene duplication, and both belong to the same topological-associating domain (TAD) [19], their expression is most likely co-regulated and dependent on the tissue-specific 3D chromatin architecture. The 3D chromatin architecture tightly regulates gene expression in time and space through physical interactions involving promoters and regulatory elements located in non-coding regions in their vicinity [20]. These regulatory networks are generally tissue-specific [21], and their impairment may interfere with the normal 3D chromatin architecture, leading to genes’ misexpression patterns, causing or aggravating disease [22,23].

Herein, we aimed to characterize a potential 3D chromatin architecture re-wiring process, occurring at the *CDH3*–*CDH1* loci, that may contribute to E-cadherin to P-cadherin switch in gastric cancer.

## 2. Materials and Methods

### 2.1. Samples

Gastric cancers from 43 patients were selected for the study, displaying equivalent male to female ratio, range of age at diagnosis, distribution of tumour stage, and histological type. Clinicopathological features were collected from clinical records. The present project was approved by the Ethical Committee of Centro Hospitalar Universitário de São João on 16 March 2017, with internal reference CES072017.

### 2.2. RNA Extraction

Tumour areas with at least 75% tumour cells were selected for RNA extraction using MagMax FFPE DNA/RNA Ultra Kit (Applied Biosystems, Life technologies, Carlsbad, CA, USA), according to the manufacturer’s instructions. Briefly, sections of paraffin-embedded tumour samples were de-paraffinized and protease K-digested. RNA was bonded to magnetic beads, pelleted against a magnetic stand, and the supernatant containing RNA washed and eluted. Concentrations were determined using Qubit^®^ 2.0 Fluorometer (Invitrogen, Life Technologies Europe BV, Porto, Portugal) with RNA high sensitivity assay, and integrity was assessed with a 2100 Bioanalyzer Instrument (Agilent Technologies, Instruments Soquimica, Lisboa, Portugal).

### 2.3. RNA-Seq and Whole Transcriptome Sequencing Bioinformatics Analysis

RNA was sequenced with Ion AmpliSeq™ Transcriptome Human Gene Expression Panel kit in an Ion Chef™ 550 NGS instrument. We obtained sequencing reads aligned to the GRCh37 human genome using TMAP [24] and annotated with USCS GRCh37. Only canonical transcripts were selected. *CDH1* and *CDH3* mRNA expression was considered positive for a threshold of 10 TPM or higher. Processed total RNA-seq from normal stomach tissue samples ENCSR853WOM, ENCSR471RUK, and ENCSR000AFI was collected from ENCODE [25,26,27].

### 2.4. Cell Lines and Culture

MKN74 human cell line was acquired from the JCRB Cell Bank, cultured in RPMI medium (GIBCO, Fisher Scientific, Porto Salvo, Portugal) supplemented with 10% fetal bovine serum (GIBCO, Fisher Scientific, Porto Salvo, Portugal) and maintained at 37 °C and 5% CO_2_ in a high-humidity atmosphere. STR analysis was used to confirm cell identification. Cells were free from mycoplasma contamination.

### 2.5. Generation of CRISPR-Cas9 E-Cadherin Deletion

Single guided RNAs (sgRNAs) were designed to target the *CDH1* exon 2 and impair E-cadherin function, resourced using the Zhang Lab tool, currently available at Benchling online platform. Individual sgRNAs (Sigma-Aldrich, Merck Life Science, Darmstadt, Germany) (Appendix A) were cloned into pSpCas9(BB)-2A-Puro (PX459) V2.0 plasmid (Addgene 62988), in BbsI restriction site (New England Biolabs), according to the method of Ran and colleagues [28]. Each sgRNA-plasmid was transformed into stbl3 cells (Invitrogen), and colonies were screened by PCR and Sanger sequenced (Appendix A). sgRNAs were transfected in MKN74 cell line. Briefly, cells were seeded in 12-well plates and grown for 24 h, and pairs of plasmids were transfected with lipofectamine 3000 (Invitrogen), according to the manufacturer’s instructions. Puromycin (Merck, Merck Life Science, Darmstadt, Germany) treatment started at 48 h and was renewed every 72 h, until non-transfected cells were dead.

### 2.6. Genotyping of Edited Clones

gDNA was extracted using NZY Tissue gDNA Isolation kit (NZYTech, Lisboa, Portugal) according to the manufacturer’s protocol. gDNA was amplified with multiplex PCR kit (Qiagen, Germantown, MD, USA) and primers flanking the edition sites (Appendix A, Sigma-Aldrich). Edition was confirmed by Sanger, sequenced using BigDye Terminator v.3.1 cycle sequencing kit (Thermo Fisher Scientific, Waltham, MA, USA, EUA) on an ABI-3130 Genetic Analyzer (Applied Biosystems Europe B.V., Porto, Portugal).

### 2.7. mRNA and Protein Expression Analysis

*CDH1*/*3* mRNA expression was assessed by qPCR in triplicate. Briefly, RNA was extracted using mirVana RNA Isolation Kit (Invitrogen), according to manufacturer’s protocol. cDNA was synthesized using 1 µg of template RNA, and SuperScriptII reverse transcriptase (Invitrogen) was used to synthesize cDNA, according to the manufacturer’s instructions. *CDH1* mRNA expression was analysed by qPCR with KAPA PROBE FAST qPCR Master Mix (2X) Kit (Sigma-Aldrich) and probes for *CDH1* (Hs.PT.58.3324071, IDT), *CDH3* (Hs.PT.51.5028751, IDT), and *18S* (custom assay, IDT) as endogenous control. Reactions were sequenced on a 7500 Real-Time PCR System (Applied Biosystems). Relative expression was normalized for the endogenous *18S* control and quantified using the 2^−∆∆Ct^ method [29].

E- and P-cadherin expression was assessed by flow cytometry in triplicate. Cells were blocked for 30 min with 3% bovine serum albumin–phosphatase buffer saline (BSA, NZYTech) and incubated with primary mouse antibodies HECD-1 or P-cadherin (1:100 dilution; 1 h at 4 °C; Invitrogen), followed by secondary anti-mouse Alexa Fluor 647 antibody (1:500; 45 min at 4 °C; Invitrogen). FACS ARIA (BD Biosciences, Franklin Lakes, NJ, USA) was used to measure mean fluorescence, and Flow Jo version 10 software was used to analyse the data.

### 2.8. Statistical Analysis

GraphPad Prism version 7.00 software (GraphPad Software Inc, San Diego, CA, USA) was used to perform statistical analysis. Student’s *t*-test was used for comparison analysis, assuming equal variance between clones and wild-type samples. Differences were considered significant for *p*-value <0.05.

### 2.9. Immunofluorescence

Cells were fixed for 20 min in 4% paraformaldehyde (Merck Life Science, Darmstadt, Germany) and blocked with 5% BSA (NZYTech) for 30 min. Cells were incubated with primary mouse antibody anti-E-cadherin (24E10) or anti-β-catenin (1:50; ON; 4 °C, Cell Signaling Technology, Danvers, MA, EUA) and primary rabbit monoclonal antibody anti-P-cadherin (#2130; 1:50; ON; 4 °C, Cell Signaling Technology), followed by secondary antibodies Alexa Fluor 488 Donkey Anti-Mouse (1:500; 60 min; Life Technologies) or Alexa Fluor 594 anti-rabbit (1:500, 60 min; Life Technologies). Cell nuclei were stained with DAPI (1 μg/mL in PBS; 5 min incubation; Sigma, Merck Life Science, Darmstadt, Germany). All coverslips were mounted using Vectashield mounting media (Vector Laboratories, Newark, CA, USA) and cells were analysed by fluorescence microscopy (Imager.Z1, AxioCam fluorescence microscope or Eclipse TE-2000, both from Zeiss, Oberkochen, Germany) using AxioVision software (Rockville, White Plains, NY, USA).

### 2.10. Protein Extraction

Cells were lysed with lysis buffer containing 1% (*v*/*v*) Triton X-100 and 1% (*v*/*v*) IGEPAL in PBS, supplemented with phosphatase (Sigma-Aldrich) and protease inhibitor cocktails (Roche, Mannheim, Germany) at 4 °C. Modified Bradford Assay (Bio-Rad, Hercules, CA, USA) was used to quantify total protein content of the samples.

### 2.11. Proteomics Sample Processing

Proteomics was performed according to solid-phase-enhanced sample-preparation (SP3) protocol [30]. Briefly, proteins were solubilized for 10 min with 100 mM Tris pH 8.5, 1% sodium deoxycholate, 10 mM tris(2-carboxyethyl) phosphine (TCEP), and 40 mM chloroacetamide at 95 °C at 1000 rpm. Proteins were digested with Trypsin/LysC (2 μg; Thermo Fisher Scientific, Waltham, MA, USA, EUA) overnight at 37 °C and 1000 rpm. Peptides were cleaned up and quantified.

### 2.12. Protein Identification and Quantification, and Enrichment GO Terms Analysis

Proteome Discoverer 2.1 software was used for identification (UniProt/SwissProt human database using Mascot (Version 2.5.1, Matrix Sciences)) and/or quantification of proteins. Carbamidomethylation of cysteines was set as fixed modification and methionine oxidation as variable modification, with trypsin as selected enzyme. High confidence master proteins with at least two unique peptides and without contaminants were considered. Differentially expressed proteins were selected considering adjusted *p*-value ≤0.05, abundance ratio ≤0.67 (downregulated) or ≥1.5 (upregulated), and concordance between replicates. ClusterProfiler (v.4.4.4) R package was used for enrichment GO terms analysis (q-value < 0.05). Statistics were performed using R (v.4.2.0).

### 2.13. ATAC-Seq

ATAC-seq data from normal stomach tissue were collected from phase 3 ENCODE project [31]. ATAC-seq libraries were generated from MKN74 cell line and *CDH1* deletion clone, as described previously [32]. A total of 5 × 10^4^ cells were lysed and chromatin was tagmented using Tn5 transposase and purified with MiniElute PCR purification kit (Qiagen), following the manufacturer’s protocol. The appropriate number of PCR cycles was determined using SYBR Green qPCR. Transposed DNA fragments were amplified by PCR and purified using QIAquick PCR purification kit (Qiagen), following the manufacturer’s protocol. Quality control of ATAC-seq libraries was accessed by tapestation, and samples were sequenced in Illumina HiSeqX technology according to standard protocols. ATAC-seq peaks were mapped resourcing to an ENCODE-DCC ATAC-seq bioinformatics pipeline with default parameters for adaptors trimming (cutadapt: -e 0.1 -m 5), alignment (bowtie), quality filtering for mapq < 30, removal of duplicates and mitochondrial reads (piccard), peak calling (macs2), and replicate statistical analysis (idr, *p*-value < 0.05).

### 2.14. C-Seq

Fresh gastric tissue from bariatric surgeries was collected and washed in HBSS 1x (GIBCO, Fisher Scientific, Porto Salvo, Portugal). Gastric mucosa cells were scraped and enzymatically dissociated with collagenase type I (Merck Life Science, Darmstadt, Germany), dispase (Merck), trypsin inhibitor (Sigma-Aldrish, Merck Life Science, Darmstadt, Germany), BSA (Nzytech, Lisboa, Portugal), dithiothreitol (Invitrogen Life Technologies Europe BV, Porto, Portugal), and HBSS 1× (GIBCO, Fisher Scientific, Porto Salvo, Portugal) at 37 °C and 150 rpm for at least 1 h. From these, 4C-seq libraries were generated, as previously described [33,34], containing 1 × 10^7^ cells, cross-linked in 2% paraformaldehyde and lysed. Nuclei suspensions were digested with DnpII (New England Biolabs, EVRY cedex, Évry-Courcouronnes, France) as primary and Csp6I (New England Biolabs) as secondary restriction enzymes and re-circularized with T4 DNA Ligase (Thermo Fisher Scientific). Then, 4C-seq libraries were purified using Amicon Ultra-15 10 kDa (MWCO) (Millipore) and PCR-amplified with 3.2 µg per reaction (Appendix A). The *CDH1* viewpoint was designed to target chr16: 68737018-68737215, hg38. Samples were paired-end sequenced on Illumina HiSeqX technology (150 bp reads) according to standard protocols. *CDH1* interactions on a genome scale were mapped from sequenced 4C libraries, using a bioinformatics pipeline based on Pipe4C [34] and PeakC [35] with window size 2, alpha fdr 0.1, and minimal distance 500. Collection of bariatric surgeries’ material was approved by the Ethical Committee of Centro Hospitalar Universitário de São João on 8 May 2020, with internal reference CE 305-19.

## 3. Results

### 3.1. CDH1 to CDH3 mRNA Expression Switch Is a Frequent Event in Gastric Cancer

To understand the frequency of and explore potential mechanisms which underly E- to P-cadherin switching in gastric cancer, we started by analysing RNA-seq data from 43 gastric cancers to evaluate *CDH1* and *CDH3* mRNA expression. We found that *CDH1* downregulation and *CDH3* upregulation are the most frequent combination of events in this cohort, occurring in 42% of all cases, followed by expression of both cadherins (33%) (Figure 1A). Downregulation of both cadherins occurred in 19% of tumours, while only 7% of all tumours expressed *CDH1* and lacked *CDH3* expression (Figure 1A). In this cohort, 75% of the tumours expressed *CDH3* (Figure 1A, upper panels), a feature that is acquired in tumours, as normal stomach epithelial cells do not express *CDH3* (Figure 1B). In 42% of all cases, there was a *CDH1* to *CDH3* expression switch (Figure 1A, upper left panel).

### 3.2. E-Cadherin Loss of Function Triggers E-Cadherin to P-Cadherin Switch in Gastric Cancer

Given the abundance of gastric cancers bearing simultaneous loss of *CDH1* expression and *CDH3* upregulation, we tested whether depleting *CDH1* would trigger *CDH3* upregulation.

We used sgRNAs flanking *CDH1* exon 2 (Appendix A), and successful removal of exon 2 was confirmed by Sanger sequencing in the MKN74 cell line (Appendix A). This clone carried a homozygous deletion encompassing the *CDH1* exon 2 (Figure 2A) and presented complete loss of both *CDH1* mRNA and E-cadherin protein expression (Figure 2B). As a consequence of this deletion, both *CDH3* mRNA and P-cadherin protein expression increased by 60% (*p*-value = 0.05 and *p*-value = 0.037, respectively) (Figure 2C). The immunofluorescence data highlight the P-cadherin expression increase at the plasma membrane upon E-cadherin expression loss, likely rescuing adherens junctions (Figure 2D). Indeed, P-cadherin co-localized with adhesion partner β-catenin at the cell membrane in the *CDH1 deletion* clone, while the parental cell line displayed lower (and heterogeneous) P-cadherin, as was the case in the cytoplasm (Figure 2D). Altogether, these results support that depletion of E-cadherin, due to gene deletions, can trigger E- to P-cadherin switch.

### 3.3. E-Cadherin Expression Loss Triggers Downregulation of Adhesion Complex Partners and Adhesion-Related Pathways

To further dissect the impact of E-cadherin loss, we used proteomics to evaluate the differential proteome between the parental cell line and the *CDH1* deletion clone. A total of 4579 proteins were identified, from which 41 were downregulated and 39 were upregulated in the *CDH1* deletion clone (Appendix A). The E- to P-cadherin switch was confirmed in the *CDH1* deletion clone, with downregulation of the former (0.37-fold, *p*-value = 0.0002) and upregulation of the latter (1.99-fold, *p*-value = 0.018) (Figure 2D). In addition to E- and P-cadherin expression, other adhesion complex partners became impaired in the *CDH1* deletion clone, namely, *CTNNA1* (αE-catenin) and *CTNNB1* (β-catenin), while others remain unchanged, namely, *CTNND1* (p120) (Figure 2D). Gene ontology analysis revealed several adhesion-associated pathways downregulated upon *CDH1* deletion (Figure 2E, Appendix A), encompassing ‘adherens junction’, ‘cadherin binding’, ‘focal adhesion assembly’, and catenin- and actin-associated pathways (Figure 2E, Appendix A). In contrast, the ‘establishment and maintenance of cell polarity’ pathway became upregulated, likely revealing the rescue by P-cadherin. Events associated with migration, such as ‘cell projection membrane’ and ‘wound healing’, were found upregulated in cells with E- to P-cadherin switch (Figure 2E, Appendix A). Downregulation of immune-associated pathways ‘macrophage chemotaxis’ and ‘regulation of leucocyte migration’ and upregulation of oncogenic pathway ‘Ras protein signal transduction’ were also found upon *CDH1* deletion (Figure 2E, Appendix A).

### 3.4. D Chromatin Architecture Re-Wiring at the CDH3–CDH1 Loci Contributes to E-Cadherin to P-Cadherin Switch in Gastric Cancer 

In an attempt to identify the mechanism triggering the E- to P-cadherin switch, upon *CDH1* depletion, we analysed regulatory elements within the *CDH3*–*CDH1* loci that may control expression of cadherins in the stomach. We mined the Ensembl database and found an eQTL located within the *CDH3* intron 2 (chr16: 68661867-68662578 and chr16: 68663306-68670017, hg38) (Figure 3A). Normal epithelial cells from bariatric surgeries, MKN74 parental cells as well as *CDH1* deletion cells, were profiled for *CDH1*-promoter interactions with the eQTL using 4C-seq, and for accessibility of chromatin using ATAC-seq. Normal stomach accessible chromatin was also collected from the ENCODE project and analysed [31]. In normal stomach, which expresses *CDH1* but not *CDH3*, the chromatin was not accessible at the *CDH3* promoter nor at the *CDH3*-eQTL, but was accessible at the *CDH1* promoter (Figure 3A, ATAC-seq normal stomach). Neither the *CDH3* promoter nor the *CDH3*–eQTL interacted with the *CDH1* promoter (Figure 3A, 4C-seq normal stomach). In contrast, upon *CDH1* deletion, availability of the chromatin increased around the eQTL sequence and was maintained as available at the *CDH1* and *CDH3* promoters (Figure 3A, ATAC-seq parental vs. ATAC-seq *CDH1* deletion). Additionally, new interactions were established between the *CDH1* promoter and the *CDH3*–eQTL region (Figure 3A, 4C-seq parental vs. 4C-seq *CDH1* deletion). These chromatin architecture changes seem to indicate that, upon *CDH1*/E-cadherin depletion, the *CDH1* promoter interacts with the *CDH3* eQTL, potentially leading to *CDH3*/P-cadherin expression increase (Figure 2B–D). If this is the case, deleting the *CDH3*–eQTL in a cell line expressing both *CDH3* and *CDH1* could provide information on the expression regulatory role of the *CDH3*-eQTL. To address this, we CRISPR-Cas9 deleted the eQTL in parental MKN74 cells. This led to a 50% decrease in both *CDH3* and *CDH1* mRNA expression (*p*-value = 0.0196 and *p*-value = 0.0084, respectively), and a 30% decrease in P-cadherin and E-cadherin protein expression (*p*-value = 0.035 and *p*-value = 0.0486, respectively) as compared to parental MKN74 cells (Figure 3B). These results pinpoint *CDH3*–eQTL as a potential enhancer of both *CDH3* and *CDH1* in this cell line. Altogether, these data provide evidence that loss of *CDH1*/E-cadherin expression alters the chromatin conformation at the *CDH3* locus, allowing interaction of the *CDH1* promoter with the *CDH3*–eQTL, which seems to potentiate *CDH3*/P-cadherin expression.

## 4. Discussion

Herein, we aimed to characterize a potential 3D chromatin architecture re-wiring process, occurring at the *CDH3*–*CDH1* loci, that may explain the *CDH1*/E-cadherin to *CDH3*/P-cadherin switch observed in gastric cancer. First, we demonstrated that 75% of the gastric cancers analysed overexpressed *CDH3*. Additionally, we observed that, in 42% of all cases, *CDH3* overexpression occurred concomitantly with *CDH1* downregulation, representing an E- to P-cadherin switch. We also found that an eQTL within the *CDH3* intron 2, acting as a regulatory element, was able to modulate both *CDH1* and *CDH3* expression, and that it interacted with the *CDH1* promoter upon *CDH1* loss of expression. To the best of our knowledge, this body of results highlights a novel mechanism which triggers E- to P-cadherin switch in gastric cancer.

Our data suggest that E- to P-cadherin switch is a common event in gastric cancers. The impact of P-cadherin overexpression in cancer is still controversial, as this molecule seems to act as tumour-promoting and a marker of a bad prognosis in several cancer types—namely, in breast, glioblastoma, and gastric cancer [36]—but may also act as a marker of a good prognosis—namely, in colon cancer [37]. Breast cancer cells presenting E- to P-cadherin switch, and those expressing both cadherins, reveal increased migration capacity in vitro [38], while breast cancer epithelial cells expressing E-cadherin and lacking P-cadherin expression, resembling the normal breast, and cells lacking both cadherins display reduced migration [38].

Herein, we created a model of E- to P-cadherin switch by using CRISPR-Cas9 editing to deplete *CDH1*. This model presented reduced expression of adhesion complex partners, but maintained β-catenin expression at adherens junctions, which co-localized with P-cadherin at the cell membrane. This likely represents an adherens junctions rescue by P-cadherin in E-cadherin-depleted cells, as previously reported [39]. Evidence for E- to P-cadherin switch has been linked to invasive lobular breast cancer with tubular elements, which show reduced E-cadherin expression, normal β-catenin expression, and P-cadherin upregulation, highlighting rescue of adherens junctions in breast tubular elements when E-cadherin is lost [39].

Our data on differential protein expression and gene ontology analysis in the *CDH1* deletion clone, in comparison to an isogenic parental *CDH1* cell line, highlighted downregulation of adhesion-, catenin- and actin-related pathways, as well as upregulation of cell polarity, all associated with E-cadherin main functions. Despite the fact that E-cadherin deficiency weakens cell-adhesion, cell polarity regulation further supports P-cadherin rescue of adherens junctions (Christgen et al. 2020). Our gene ontology data also support increased migration potential of cells with E- to P-cadherin switch, namely, through activation of the ‘wound healing’ pathway, also described for breast cancer cells [38]. Our findings also suggest that cells with E- to P-cadherin switch may have increased proliferation through the activation of Ras oncogenic signalling pathways. As in many published reports, the abovementioned findings support that E- to P-cadherin switch is associated with more aggressive tumour phenotypes.

In the present study, we used the abovementioned cell model to identify the mechanism responsible for the E- to P-cadherin switch. We raised the hypothesis that regulatory elements in the *CDH3*/*CDH1* loci could play a role in the observed E- to P-cadherin expression switch, as gene expression is tightly regulated through the physical proximity enabled by the 3D-chromatin architecture (Andrey and Mundlos 2017) [20]. We demonstrated that, upon *CDH1* depletion triggered by a gene deletion, the chromatin re-shuffles at the *CDH3*/*CDH1* loci, and its promoter becomes physically bound to an eQTL overlapping accessible chromatin located in *CDH3* intron 2; this associates with increased *CDH3*/P-cadherin expression. The absence of interactions between the *CDH1* promoter and this eQTL in normal stomach epithelial tissue may explain why *CDH3* is not expressed in this particular tissue. While these data provide a rationale for E- to P-cadherin switch in gastric cancer, they do not inform whether the eQTL promotes *CDH3* expression. To address this question, we deleted the eQTL in a tumour cell line expressing wild-type E- and P-cadherins, preventing it from interacting with the *CDH1* promoter. This resulted in downregulation of both P- and E-cadherin expression and demonstrated that the *CDH3*–eQTL controlled P-cadherin expression. Taken together, these data provide evidence that loss of *CDH1*/E-cadherin expression alters the chromatin conformation at the *CDH3* locus, allowing interaction of the *CDH1* promoter with the *CDH3*–eQTL, which seems to potentiate *CDH3*/P-cadherin expression. Our findings also suggest that the *CDH3*–eQTL acts as an enhancer regulating both P- and E-cadherin protein expression, thus regulating P-cadherin expression in the switch mechanism. Similar findings have been observed for the ZRS enhancer located within the *Lmbr1* gene, also controlling *Shh* gene expression [40].

Single-cell sorting of CRISPR-Cas9 clones herein described only yielded one homozygous clone for each set of sgRNAs harbouring the intended deletions, which we acknowledge as a study limitation.

## 5. Conclusions

Overall, this study provides insights into a noncoding regulatory element that controls E- and P-cadherin expression and triggers E- to P-cadherin switch in the presence of a *CDH1* depletion, through a re-shuffle of the *CDH1*/*CDH3* loci.

## Figures and Tables

**Figure 1 biology-12-00803-f001:**
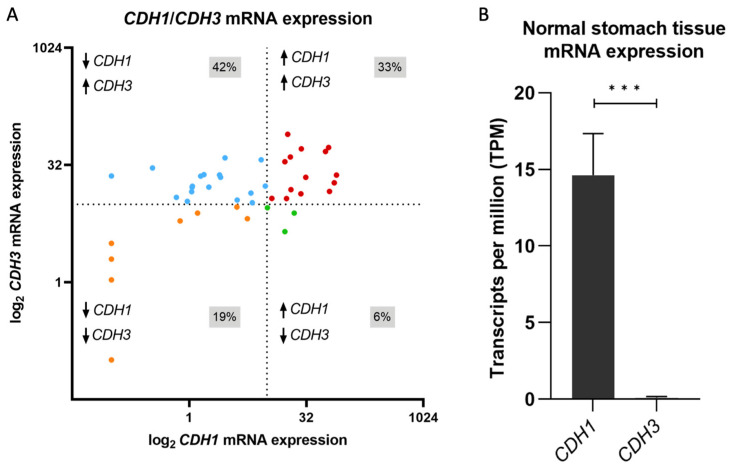
*CDH1* and *CDH3* mRNA expression in gastric tissues. (**A**) Distribution of *CDH1* and *CDH3* mRNA expression in gastric cancer tumours. Blue dots represent gastric cancers with low *CDH1* and high *CDH3* expression, red dots represent gastric cancers with high expression for both *CDH1* and *CDH3*, orange dots represent gastric cancers with low *CDH1* and *CDH3* expression, green dots represent gastric cancers with high *CDH1* and low *CDH3* expression. Dashed lines represent 10 TPM, the threshold for low or high expression. Data shown in transcripts per million (TPM). (**B**) *CDH1* and *CDH3* mRNA expression in stomach normal tissue from ENCODE [25,26,27]. Data shown in transcripts per million (TPM) (*** *p* value ≤ 0.001).

**Figure 2 biology-12-00803-f002:**
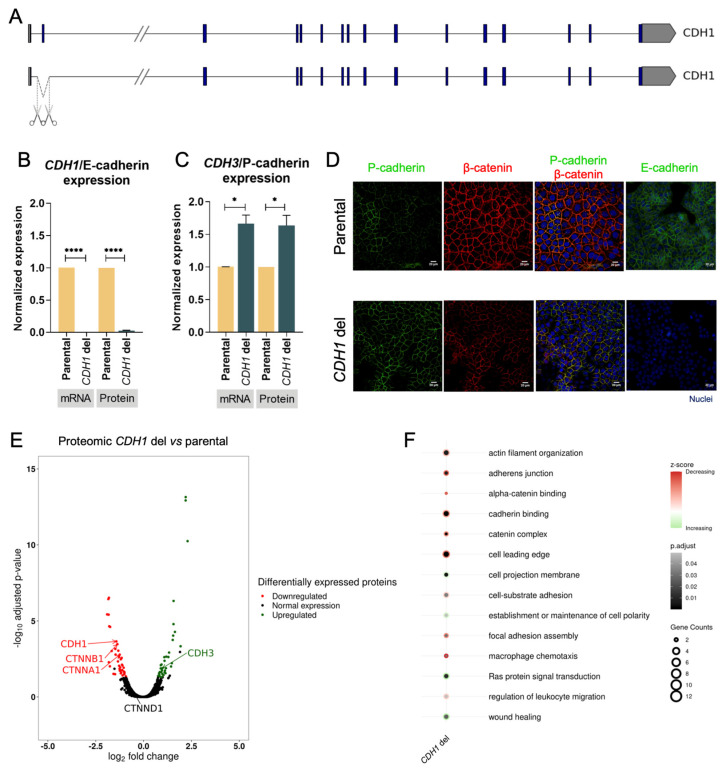
Characterization of E- to P-cadherin switch. (**A**) CRISPR-Cas9 design strategy. (**B**) Characterization of *CDH1* deletion in *CDH1*/E-cadherin mRNA and protein expression. Values represented as mean ± SEM. Protein expression represented as median fluorescence intensity (MFI) (**** *p* value ≤ 0.0001). (**C**) Characterization of *CDH1* deletion in *CDH3*/P-cadherin mRNA and protein expression. Values represented as mean ± SEM. Protein expression represented as median fluorescence intensity (MFI) (* *p* value ≤ 0.05). (**D**) Immunofluorescence of parental cells (top) and *CDH1* deletion cells (down) for P-cadherin, β-catenin, and E-cadherin. Nuclei were stained with DAPI (blue), and white scale bars represent a distance of 20 μm. (**E**) Volcano plot illustrating differentially expressed proteins in *CDH1* deletion vs. parental cells. (**F**) Enrichment analysis of biological processes, molecular function, and cellular components in upregulated and downregulated proteins.

**Figure 3 biology-12-00803-f003:**
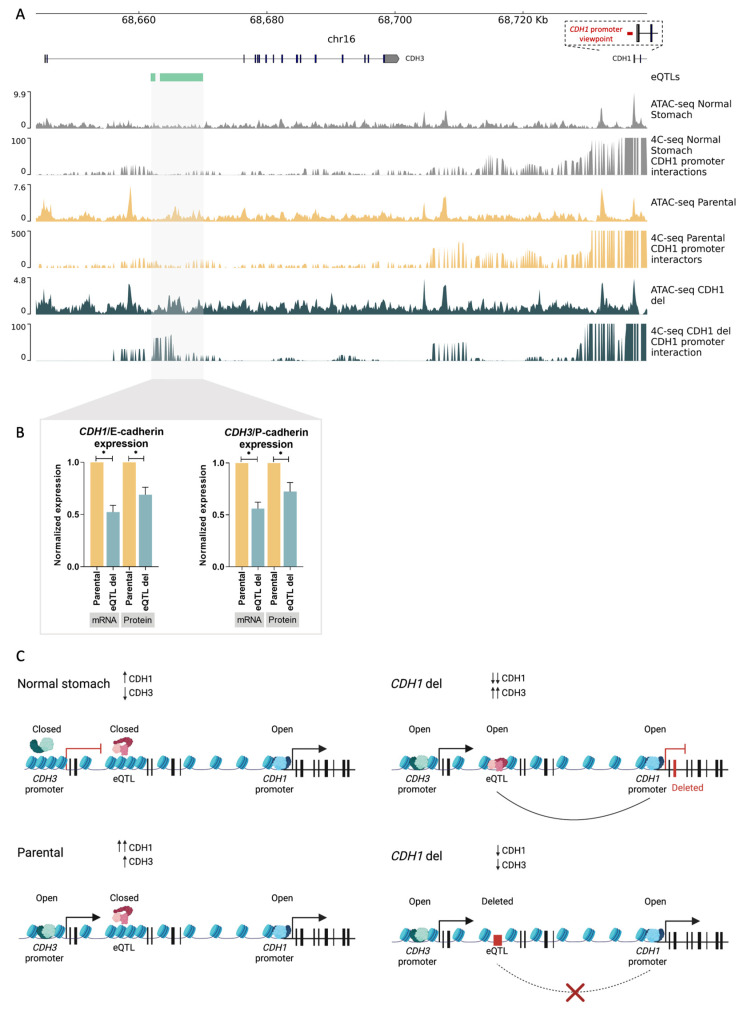
The 3D chromatin architecture in E- to P-cadherin switch. (**A**) Accessible chromatin and *CDH1*-promoter interactors in normal stomach tissue, parental MKN74 cells, and *CDH1* deletion clone. *CDH1* promoter viewpoint is shown in red. (**B**) Characterization of *CDH3*–eQTL deletion in *CDH1*/E-cadherin and *CDH3*/P-cadherin mRNA, and protein expression. Values represented as mean ± SEM. Protein expression represented as median fluorescence intensity (MFI) (* *p* value ≤ 0.05). (**C**) Model of the role of 3D chromatin architecture in E- to P-cadherin switch. Red rectangles represent deletions.

## Data Availability

Raw data supporting the findings of this study are available upon request from the corresponding author. Processed data are available in Appendix A.

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
