# Peer review of "3D Chromatin Architecture Re-Wiring at the CDH3/CDH1 Loci Contributes to E-Cadherin to P-Cadherin Expression Switch in Gastric Cancer"

_biology, 2023, doi:10.3390/biology12060803_

Round 1

Reviewer 1 Report

In the manuscript, Biology-2380641, José et al reported their work on the role of the local 3D chromatin structure in the regulation of gene expression during the E-cadherin to P-cadherin expression switch in gastric cancer. The authors found (i) that CDH1 to CDH3 expression switch occurred in 42% of gastric tumors analyzed; (ii) that CDH1 led to increased CDH3/P-cadherin expression at plasma membrane; (iii) that E- to P-cadherin switch was accompanied with increased CDH1 promoter interactions with CDH3-eQTL, which is absent in normal stomach and parental cells; and (iv) CDH3-eQTL deletion resulted in reduced expression of CDH3/CDH1.

Major concerns:

1.     In the two CRISPR deletion analyses (CDH1 knockdown and CDH3-eQTL deletion), only a single cell clone with a single sgRNA pair was characterized in each case. To ensure the reproducibility of the results (in Figs 2 and 3), it is necessary to conduct the assay in at least two cell clones.

2.     Fig 1B showed that, in normal stomach cells, CDH3 expression is near zero. However, Fig 1A showed that about 22% tumor cells has reduced CDH3 expression --- compared to what cells?

3.     In Fig 3A, please indicate the viewpoint of the 4C-seq tracks.

Minor editing of English language required.

Author Response

Cover Letter and Point-by-point reply to Reviewers

Porto, 25th May, 2023

Manuscript ID: biology-2380641

Revised Title: “3D chromatin architecture re-wiring at the CDH3/CDH1 loci contribute to E-cadherin to P-cadherin expression switch in gastric cancer”

Dear Professors Jukka Finne and Andrés Moya, Editors-in-Chief of Biology, and Ilda Ribeiro, Isabel Carreira and Alberto Órfão, guest editors of New Sight in Cancer Genetics special issue, Dear Reviewers,

We greatly appreciate this comprehensive peer review of our manuscript that allowed us to improve the clarity and quality of our paper.

We herein submit an updated version of the manuscript taking into consideration all the comments from Reviewers 1, 2 and 3. We submit a clean version with each alteration highlighted as a comment on the side of the text, numbered according to the Reviewers questions. In this complementary file, we list all comments, which were answered point-by-point.

We hope that the Editor-in-Chief of Biology, Reviewers and the Editorial Manager find these amendments to our manuscript adequate and sufficient to proceed with the publication process.

Sincerely,

Carla Oliveira

Reply to Reviewer 1 comments and suggestions:

1) In the two CRISPR deletion analyses (CDH1 knockdown and CDH3-eQTL deletion), only a single cell clone with a single sgRNA pair was characterized in each case. To ensure the reproducibility of the results (in Figs 2 and 3), it is necessary to conduct the assay in at least two cell clones.

We thank and agree with Reviewer 1 observation. Indeed, CRISPR-Cas9 editing, clones genotyping and characterization is a very laborious experiment. Unfortunately, from two 96-well plates, we were only able to obtain one homozygous clone from each set of sgRNAs harbouring the intended deletions in the single cell sorting experiment. We hope the reviewer understands this limitation, which we have added to the discussion.

Page 11, line 410-412: “Single cell sorting of CRISPR-Cas9 clones herein described only yielded one homozygous clone for each set of sgRNAs harbouring the intended deletions, which we acknowledge as a study limitation.”

2) Fig 1B showed that, in normal stomach cells, CDH3 expression is near zero. However, Fig 1A showed that about 22% tumor cells has reduced CDH3 expression --- compared to what cells?

We thank Reviewer 1 for this comment, that we agreed with. Figure 1B shows CDH3 expression in normal stomach tissue of ENCODE samples. Therefore, we edited the materials and methods section and Figure 1 legend to include this mention.

Page 3, line 116-118: “Processed total RNA-seq from normal stomach tissue samples ENCSR853WOM, ENCSR471RUK, ENCSR000AFI was collected from ENCODE (Feingold et al. 2004; Zhang et al. 2019; Lee et al. 2020).”

Page 6, line 252-253: “B. CDH1 and CDH3 mRNA expression in stomach normal tissue from ENCODE (Feingold et al. 2004; Zhang et al. 2019; Lee et al. 2020). Data shown in transcripts per million (TPM).”

Indeed, CDH3 expression is extremely low in normal stomach tissue and for that reason is not visible in figure 1B. We would like to provide an explanation for the fact that in our cohort we considered low or absent CDH3 expression for a threshold of 10 TPM, value for which CDH3 expression was not exceeded in all samples of normal stomach tissue. This feature was stated in the materials and methods and represented with dash lines for both genes in Figure 1A. Thus, we edited the Figure 1 legend to include this mention. Figure 1A results are shown in log2, thus we edited the axis title to include tis mention as well.

Page 3, line 115-116: “CDH1 and CDH3 mRNA expression was considered positive for a threshold of 10 TPM or higher.”

Page 6, line 250-251: “Dashed lines represent 10 TPM, the threshold for low or high expression.”

Tumour samples results are not compared to any sample, but represented as TPM, thus, we edited the figure legend for clarification, as below.

Page 6, line 247-250: “Blue dots represent gastric cancers with low CDH1 and high CDH3 expression, red dots represent gastric cancers with high expression for both CDH1 and CDH3, orange dots represent gastric cancers with low CDH1 and CDH3 expression, green dots represent gastric cancers with high CDH1 and low CDH3 expression.”

Therefore, tumour cells that with low CDH3 expression, do not differ from normal stomach tissue, which lacks CDH3expression.

3) In Fig 3A, please indicate the viewpoint of the 4C-seq tracks.

We acknowledge and agree with Reviewer 1 observation. Therefore, we edited figure 3 to include a zoom showing the viewpoint in the CDH1 promoter and added this mention to the figure legend and added the coordinates in the materials and methods section.

Page 6, line 340: “CDH1 promoter viewpoint is shown in red.”

Page 5, line 225: “The CDH1 viewpoint was designed to target chr16: 68737018-68737215, hg38.”

Reply to Reviewer 3 comments and suggestions:

1) Although this study highlights a triggering E- to P-cadherin switch in gastric cancer, the reviewer is not sure if these can lead to 3D chromatin architecture. The study does not include a single 3D experiment for chromatin architecture and/organization.

To get such conclusion, 3D studies experiment such as 3D FISH for both genes are required.

Or changing the conclusion and title avoid mentioning about 3D chromatin architecture.

We acknowledge Reviewer 3 observations, and we really appreciated this suggestion to improve our manuscript. We would like to call the attention for section 13 and 14 of the materials and methods, in which 4C-seq and ATAC-seq were performed in normal stomach from bariatric surgeries, MKN74 parental cell line and Exon2 del isogenic clone. We hope the reviewer agrees that any of the above experimental approaches fall into the 3D chromatin architecture assays’ category. Further, the changes observed in the 4C-seq experiments reveals changes in chromatin interaction, which are part of the 3D genome architecture.

2) Line 198, 14.4. C-seq, should be 14

We thank Reviewer 3 for this observation and revisited the typing error.

Page 5, line 214: “14. 4C-seq”

3) Line 284, the D is a typo (D chromatin architecture re-wiring)

We thank Reviewer 3 for this observation and revisited the typing error.

Page 8, line 304-305: “4. 3D chromatin architecture re-wiring at the CDH3-CDH1 loci contributes to E-cadherin to P-cadherin switch in gastric cancer”

4) The resolution of tags and plots in figure 2 and 3 should be improved

We acknowledge and agree with Reviewer 3 comment, therefore we increased the resolution of tags and plots in all figures.

5) The authors did not provide the number of their ethical approval.

We acknowledge and agree with Reviewer 3 comment, for which we revised the manuscript in the materials and methods and ethical approval sections.

Page 2, line 99-101: “The present project has been approved by the Ethical Committee of Centro Hospitalar Universitário de São João on March 16th, 2017 with internal reference CES072017.”

Page 5, line 230-232: “Collection of bariatric surgeries’ material has been approved by the Ethical Committee of Centro Hospitalar Universitário de São João on May 8th, 2020 with internal reference CE 305-19.”

6) I could not find this page (http://CRISPR.mit.edu.) that authors used to generate CRISPR-Cas9 E-cadherin deletion.

We thank and agree with Reviewer 1 observation. We would like to provide an explanation for the fact that this tool was used in 2016 to design the single guided RNAs used in this study. However, it is not currently available online at this link, but within the benchling online platform. Therefore, we made the following manuscript correction.

Page 3, line 125-127: “Single guided RNAs (sgRNAs) were designed to target CDH1 exon 2 and impair E-cadherin function, resourcing to Zheng lab tool, currently available at benchling online platform.

7) Most of the sub-sections are missing a citation, for example RNA-seq and whole transcriptome sequencing bioinformatics analysis, Generation of CRISPR-Cas9 E-cadherin deletion, mRNA and protein expression analysis, Statistical analysis, Label free proteomic quantification: Hughes et al. 2019 should be indexed, Protein identification and quantification and enrichment GO terms analysis, ATAC-seq ; Buenrostro et al. 2015  shouod be indexed

We revisited the text according to Reviewer 2 comments, that we appreciate, and added the missing citations.

Page 3, line 113-114: “We obtained sequencing reads aligned to the GRCh37 human genome using TMAP (Tennekes 2018)

Page 3, line 129: “according to Ran and colleagues (Ran et al. 2013).”

Page 4, line 150-152: “Relative expression was normalized for the endogenous 18S control and quantified using the 2−∆∆Ct method (Livak and Schmittgen 2001).”

Hughes et al. 2019 and Buenrostro et al. 2015 references were indexed to the library.

Reviewer 2 Report

The data of this study provide evidence supporting that loss of CDH1/E-cadherin expression alters the CDH3 locus chromatin conformation, allowing a CDH1 promoter interaction with a CDH3 eQTL, and promoting CDH3/P-cadherin expression. These data highlight a novel mechanism triggering E- to P-cadherin switch in gastric cancer.

Overall, this study provides insights into a noncoding regulatory element that con- trols E- and P-cadherin expression and triggers E- to P-cadherin switch in the presence of a CDH1 depletion, through a re-shuffle of the CDH1/CDH3 loci.

This is an interesting study reporting the development of a genetic model to study the mechanisms of E-Cadherin and P-Cadherin expression occurring in gastric cancer.

Author Response

(The authors gave the same response as above.)

Reviewer 3 Report

The study by José et al aimed at characterizing a 3D chromatin architecture re-wiring process, occurring at the CDH3-CDH1 loci, in gastric cancers samples from 43 patients. The authors argue  that their observation might explain the CDH1/E-cadherin to  CDH3/P-cadherin switch. Although the study seems interesting, but there some serious issues should be addressed.

Major comments

-  Although this study highlights a triggering E- to P-cadherin switch in gastric cancer, the reviewer is not sure if these can lead to 3D chromatin architecture. The study does not include a single 3D experiment for chromatin architecture and/organization.

To get such conclusion, 3D studies experiment such as 3D FISH for both genes are required.

Or changing the conclusion and title avoid mentioning about  3D chromatin architecture.

Minor comments

-  There are many places where the indexing is wrong

For example

- Line 198, 14.4. C-seq , should be 14

- Line 284, the D is a typo (D chromatin architecture re-wiring)

- The resolution of tags and plots in figure 2 and 3 should be improved

- In the Material s and Methods section

1- Samples

The authors did not provide the number of their ethical approval.

5-  Generation of CRISPR-Cas9 E-cadherin deletion

- I  could not find this page (http://CRISPR.mit.edu.) that authors used to generate CRISPR-Cas9 E-cadherin deletion.

- Most of the sub-sections are missing a citation, for example

3. RNA-seq and whole transcriptome sequencing bioinformatics analysis

5. Generation of CRISPR-Cas9 E-cadherin deletion

7. mRNA and protein expression analysis

8. Statistical analysis

11. Label free proteomic quantification : Hughes et al. 2019 should be indexed

12. Protein identification and quantification and enrichment GO terms analysis

13. ATAC-seq ; Buenrostro et al. 2015  shouod be indexed

14.4. C-seq  need to change to 14

- The manuscript needs careful polishing and proof reading. At a few places, the grammar is incorrect, and sentences are not understandable.

Author Response

(The authors gave the same response as above.)

Round 2

Reviewer 1 Report

This reviewer is fine with the authors' clarifications.

Reviewer 3 Report

The authors did a good job in the revised version. I am satisfied with their answers and current version.